# Silica Nanoparticles Reinforced Ionogel as Nonvolatile and Stretchable Conductors

**DOI:** 10.3390/membranes10110354

**Published:** 2020-11-19

**Authors:** Shanshan Zhang, Zhen Li, Pei Huang, Yamei Lu, Pengfei Wang

**Affiliations:** Qian Xuesen Laboratory of Space Technology, China Academy of Space Technology, Beijing 100094, China; zhangshanshan@qxslab.cn (S.Z.); huangpei@emails.bjut.edu.cn (P.H.); 18033857082@163.com (Y.L.)

**Keywords:** Ionogel, flexible conductor, silica nanoparticles

## Abstract

Ionogels combine the advantages of being conductive, stretchable, transparent and nonvolatile, which makes them suitable to be applied as conductors for flexible electronic devices. In this paper, a series of ionogels based on 1-ethyl-3-methylimidazolium ethyl-sulfate ([C_2_mim][EtSO_4_]) and polyacrylic networks were prepared. Silica nanoparticles (SNPs) were dispersed into the ionogel matrix to enhance its mechanical properties. The thermal, mechanical and electrical properties of the ionogels with various contents of crosslinking agents and SNPs were studied. The results show that a small amount of SNP doping just increases the breaking strain/stress and the nonvolatility of ionogels, as well as maintaining adequate conductivity and a high degree of transparency. Furthermore, the experimental results demonstrate that SNP-reinforced ionogels can be applied as conductors for dielectric elastomer actuators and stretchable wires, as well as for signal transmission.

## 1. Introduction

In the past decade, flexible electronic devices have attracted great research interest because they possess the advantages of light weight, high flexibility and outstanding adaptability. Owing to such properties, flexible electronic devices have showed great promise in various fields from clinical science to robotics [1,2,3,4,5,6]. Flexible conductors play an important role in transmitting electrical energy and signals [7,8]. Therefore, scientists have made an effort to develop flexible conductor materials. Carbon grease is one of the most widely used materials in dielectric elastomer sensors/actuators [9]. It exhibits good adhesive ability and can be coated on polymer substrates as a flexible electrode without the restriction of shape change. For example, Goulbourne et al. reported a kind of dielectric elastomer actuator made of polyacrylate films and carbon grease electrodes. The membrane had a pre-stretch of three and was passively inflated to various predetermined states, and it corresponded to a maximum polar strain of 28% with an applied constant voltage of 3 kV [10]. However, carbon grease exhibits poor transparency, which limits its application in optical devices. It is reported that arrays of aligned carbon nanotubes (CNTs) can serve as flexible conductors for dielectric elastomer actuators, in which large area strain has been observed [11]. However, the binder for CNTs decreases the conductivity of the resulting electrodes, leading to the reduction of the actuation strain of the dielectric elastomer actuator. Graphene has also been developed into flexible conductors [12,13]. However, the tensile strain capacity of graphene is inadequate. Damage occurs even when the loaded strain is less than 50% [14]. Indium tin oxide (ITO) exhibits the properties of large deformation and good transparency, which allows it to be widely applied in flexible optical devices [15]. Daeneke et al. reported the synthesis of flexible two-dimensional ITO on a centimeter scale using a low-temperature liquid metal printing technique to obtain bilayer samples with a transparency above 99.3% and a sheet resistance as low as 5.4 kΩ [16]. However, ITO is considered unsuitable for flexible electronics due to the high fabrication cost involved in the vacuum-based process and the lack of flexibility for future flexible electronics [17,18]. Similar to ITO, silver nanowires (AgNWs) are known to be applied as transparent, flexible and stretchable conductors [19]. Chen et al. reported flexible transparent electrodes with a sheet resistance of around 10  Ω/sq and a transmittance of around 92%, where the electrodes were prepared with water-processed silver nanowires and a polyelectrolyte [20]. However, there are intrinsic weaknesses regarding the electrical resistance and poor mechanical adhesion properties of AgNWs, which need to be further overcome for their full utilization as transparent electrodes [21]. Hydrogel is another candidate for flexible conducting. As early as 1986, Cartmell et al. reported a urethane hydrogel located in a cavity and formed in a non-conductive tray. The matrix was adapted to stick on the skin of a patient and transmitted signals between the skin and an electrode conductor supported by the base of the tray [22]. Philipp reported a pre-stretched hydrophobic elastomer that was sandwiched between two hydrogels, which functioned as ionic conductors to fabricate active noise cancelation [23].

The hydrogel material exhibits the advantages of high conductivity, stretchability, biocompatibility and low cost. However, hydrogels could dry out due to the evaporation of water, thus hydrogels cannot perform long-term service on exposure to air [24,25]. Ionogels are a kind of three-dimensional polymer that hosts ionic liquid. They have also been applied as flexible conductive materials, especially in the field of supercapacitors, batteries and fuel cells [26,27,28,29,30,31]. Chen fabricated electromechanical transducers based on ionogels which were synthesized by polymerizing acrylic acid in ionic liquid 1-ethyl-3-methylimidazolium ethylsulfate ([C_2_mim][EtSO_4_]), in which the voltage-induced areal strain could reach up to 140% [25]. Karthikeyan reported a self-charging supercapacitor using siloxane sheets as electrodes and a siloxane-based polymeric piezo-fiber separator immobilized with an ionogel electrolyte. Due to its self-charging properties, this capacitor, subjected to various levels of compressive forces, showed its ability to self-charge up to a maximum of 207 mV [32]. Wong reported an ionogel ink for 3D direct-writing to fabricate conductive structures for stretchable sensors [33]. Ionic liquids possesses high thermal stability and negligible vapor pressure, resulting in ionogels being able to retain their shape even without being stored in aqueous solution or replenished with fresh ions; furthermore, the nonvolatility of ionogels enable the prepared devices to be used in the open air [25,26,34,35,36,37,38]. In general, the stretchability of the ionogel can be enhanced by reducing the amount of the crosslinking agent. However, the decrease in the crosslinking agent content also results in low toughness and high viscosity; thus, ionogels may further lose their shape and become difficult to manipulate. It has been reported that the incorporation of silica or silica nanoparticles can improve the thermal, mechanical and electrical properties of ionogel materials, which is widely applied as ionogel electrolyte [39,40,41,42]. In this paper, in order to solve the above problems, a series of ionogels with various contents of crosslinking agents were synthesized and silica nanoparticles (SNPs) were incorporated to increase the mechanical properties. The optimized SNP-reinforced ionogels were tested as flexible conductors for electronic devices.

## 2. Materials and Methods

### 2.1. Materials

Acrylic acid (AA), 1-ethyl-3-methylimidazolium ethyl-sulfate ([C_2_mim][EtSO_4_]) and α-ketoglutaric acid were purchased from J&K company (Beijing, China). Poly(ethylene glycol) diacrylate (PEGDA), TEOS and NH_4_OH solution were purchased from Alading company (Shanghai, China). Ethyl alcohol was purchased from Bei Jing Tong Guang Fine Chemicals Company (Beijing, China). AA was applied as monomer of the polymer network, PEGDA was applied as crosslinker and α-ketoglutaric acid acted as an initiator, as shown in Appendix A.

### 2.2. Sample Preparation

The silica nanoparticles were prepared according to the classical Stöber method [43,44]. First, 1.9 g aqueous ammonia solution containing 28% NH_4_OH was diluted in ethanol (96%) to 20 mL. Then, 8.33 g TEOS was dissolved in technical ethanol to 20 mL and was slowly added into the NH_4_OH solution. Then the mixture was stirred for 12 h at room temperature. The stirred mixture was centrifuged at 5500 r/min for 5 min and washed with deionized water in ultrasonic cleaner for 5 min. Neutral solution was obtained after centrifuging and washing four times. Freezing the neutral solution at −18 °C for 6 h and then freeze drying it at −40 °C and 40 kPa for 24 h, SNPs were finally obtained.

The preparation process of SNP-reinforced ionogel is shown in Figure 1. As a standard reaction addition, 2 mmol (0.1441 g) AA, 0.0065 g (0.8 mol% of AA) poly (ethylene glycol) diacrylate and 0.0029 g of α-ketoglutaric (2 wt% of AA) acid and SNPs (various percentages) were dissolved into 2 mL [C_2_mim][EtSO_4_]. The mixture was sequentially stirred for 30 min, sonicated for 5 min and stirred again for 5 min to form a uniform solution. Then the suspension was dripped into a polytetrafluoroethylene (PTFE)-plated mold (50 mm × 50 mm × 0.6 mm) and covered with a 1-mm-thick glass plate. Ionogel was obtained after 4 h of reaction in the UV crosslinking apparatus. Similar to the abovementioned method, a series of ionogels with different contents of crosslinking agents (1.2%–2.4% mol% of AA) and SNP (0.5 wt%, 1 wt%, 2 wt%, 3 wt%, 4 wt%) were prepared. A series of SNP-reinforced ionogels were prepared by dissolving AA, poly(ethylene glycol) diacrylate, α-ketoglutaric acid and SNPs into 2 mL of ionic liquid [C_2_mim][EtSO_4_].

### 2.3. Characterization

#### 2.3.1. The size of SNPs

The size of SNPs was measured by means of a scanning electron microscope (SEM) (SU-8010, Hitachi, Tokyo, Japan) and dynamic light scattering (DLS) (VASCO, Malvern Instruments Ltd., Malvern Worcestershire, USA). The SNPs were dispersed into water at concentration of 1 mg/mL. Ten microliters dropped on silica slice and naturally dried for SEM analysis and the remaining suspension was used for DLS analysis. The test results are shown in Appendix A. The size of the SNPs was about 540 nm with a polydispersity of 0.086.

#### 2.3.2. Transmittance

Transmittance of the samples was measured by UV-visible absorption spectra on a Perkin–Elmer Lambda (750 UV/Vis/NIR-spectrometer, Waltham, MA, USA). Samples were cut into a circle shape (3 cm diameter, 0.6 mm thick) and were formed on the test stage in light path.

Ionogels were cut into samples with dimensions of 20 mm × 20 mm and were attached to the printed alphabetic patterns. The transparencies of SNP-reinforced ionogels with different proportions were compared. Ionogels (10 mm × 10 mm × 0.6mm) were placed on glass slides, covered with cover glasses and tested with an optical microscope to observe the dispersibility of SNPs in the ionogel network.

#### 2.3.3. Mechanical Characterization

A tensile strength test was carried out on a dynamic mechanical analyzer (DMA) (TA-800, TA Instrument Ltd., Eden Prairie, MN, USA). Samples (20 mm × 7 mm × 0.6 mm) were stretched at a speed of 0.5 mm/s at room temperature and the stress and strain data were recorded.

#### 2.3.4. Electrical Characterization

The resistivity was measured via a four-point probe resistivity tester (KDY-1, Jinyang Wanda Technology Ltd., Beijing, China). The samples were cut to 1 cm × 1 cm with a thickness of 0.6 mm. Collected data was used to analyze electrical properties of the prepared ionogels.

#### 2.3.5. Stability of Ionogels in Open Air

Ionogels (10 mm × 10 mm × 0.6 mm) with various SNP contents, loaded with glass sides, were put into an oven at 100 °C for 36 h. The changes of mass fractions were recorded to evaluate the stability of ionogels.

#### 2.3.6. Demonstration of Potential Application

Square acrylic frames with a length of 50 mm were cut by means of a laser cutting machine. SNP-reinforced ionogels with a thickness of 0.6 mm were cut into round samples with a diameter of 35 mm. As shown in (Appendix A), the 1-mm-thick dielectric elastomer (VHB) was fixed on the instrument, biaxially stretched for four times and the redundant VHB was fixed and trimmed with the acrylic frame. Thus, a 4-fold pre-stretched VHB element was obtained. As shown in Appendix A, the samples of SNP-reinforced ionogels were attached with two sides of VHB and connected with the electrodes by two SNP-reinforced ionogel wires with a width of 4 mm, forming a transparent flexible actuator. As shown in Appendix A, the conductive carbon grease was coated on the both sides of a 4-fold pre-stretched VHB surface, connected to the electrodes by conductive tapes to form a flexible actuator based on conductive carbon grease.

Transparent PTFE tubes with an inner diameter of 3 mm were used as molds to prepare SNP-reinforced ionogel composite wires. They then served as stretchable cables to connect to a LED circuit and the signal line to a loud speaker, respectively.

The acrylic board of 100 mm × 100 mm with pattern was prepared by means of a laser cutting machine and the series circuit of light emitting diodes was fixed on a model, connecting the power source by means for a copper conductor. Finally, we cut the copper conductor and inserted the SNP-reinforced ionogel composite wires into the circuit (as shown in Appendix A).

The signal lines of a small loudspeaker were cut off and re-connected by stretchable cables based on SNP-reinforced ionogels. Connecting the power supply of the loudspeaker with the signal source, the conductive function of stretchable cables based on SNP-reinforced ionogels was tested (as shown in Appendix A).

## 3. Results and Discussion

### 3.1. Mechanical Properties

The mechanical properties of the ionogel samples with different contents of crosslinking agents were investigated by means of a tensile test. The stress-strain curves are shown in Figure 2a and the correlation of loaded strain with the content of crosslinking agents is shown in Figure 2b. The results show that the ionogel exhibits a large breaking elongation (about 500%) and poor strength when the crosslinking agent is 0.8 mol% of AA monomer. The breaking stress is only about 7 kPa, which indicates that the material is easy to damage during manipulation. Increasing the content of crosslinking agent is effective to enhance the mechanical strength, while decreasing the breaking strain. Other samples with a higher content of crosslinking agents can only withstand 300% elongation. Further reduction of crosslinking agent would result in the material being unable to be cured or firmed. Therefore, there is not much room for improving mechanical properties by merely adjusting the crosslinking agent. Other methods of material optimization need to be urgently introduced.

SNP were incorporated into ionogels to improve the mechanical strength of ionogels without sacrificing the stretchability. The mechanical properties of these SNP-reinforced ionogels were measured and the stress-strain curves were plotted, as shown in Figure 3a. The connection of the loaded strains with the contents of crosslinking agents of these SNP-reinforced ionogels is shown in Figure 3b. Results show that the breaking strength of ionogels has been enhanced to varying degrees after being composited with SNPs. Among them, compositing 0.5 wt% SNPs into the ionogel not only enhanced the breaking strength, but also increased the breaking strain of the ionogels. It can be seen that the breaking stress of the ionogel is improved when composited with SNPs. Compared with other samples, both the breaking stress and the breaking strain of the ionogels composited with 0.5 wt% of SNPs are improved the most significantly. A quantity of 0.5 wt% SNPs was selected to reinforce ionogels for the further fabrication of flexible electronic devices.

### 3.2. Electrical Conductivity

Ionogels are widely used as conductive materials for flexible electronic devices. Therefore, the electrical conductivity of a device is important to evaluate its application value. The conductivity of blank and SNP-reinforced ionogel samples were measured by means of a four-point probe method. There were no significant changes in the conductivity of ionogels with different contents of crosslinking agents (as shown in Appendix A). Compared with the blank ionogel samples, the conductivity of SNP-reinforced ionogels exhibit a slight decrease. With the addition of 1 wt% SNP, the reinforced ionogels showed about 1.09 S/m, as can be seen in Table 1. With increasing SNP content, the electro-conductivity no longer showed a notable change. Even taking experimental error into consideration, it can be concluded that the SNP-reinforced ionogels exhibit similar electro-conductivity to blank material, which makes them possible to be applied in flexible electronic devices. It is worth mentioning that the ionogels with 0.5 wt% SNP content, which are used for further demonstration, performed at about 1.23 S/m of electroconductivity.

### 3.3. Thermal Nonvolatility

Nonvolatility is important when ionogels are applied as flexible conductors in open air, especially in high temperature environments. The nonvolatility performance of SNP-reinforced ionogels was tested under 100 °C. The curves of mass variation versus time of the resultant samples were plotted. As can be seen in Figure 4, the mass of all samples decreased obviously at 100 °C and became relatively stable after 10 h. For the blank sample, the weight loss reached 19% after thermal treatment. In comparison, all the SNP-reinforced samples only showed 13%–15% weight losses, showing that the incorporation of SNPs enhances the thermal nonvolatility of ionogels. It is worth mentioning that the samples with a different SNP content did not show a consistent dependence of thermal stability on SNP content. This phenomenon may be caused by experimental error, since the weight changes of samples (about 50 mg) were slight. Nevertheless, the difference of the thermal stability performances between blank and SNP samples is notable as it shows the effect of SNP incorporation on thermal nonvolatility.

### 3.4. Transparency Test

At the wavelength of 250–800 nm, the transmittance of the SNP-reinforced ionogels is shown in Figure 5a. We found that an increase in SNP content leads to an increase in transmittance. A possible explanation of this phenomenon is that SNPs do not absorb short-wave visible light as much as ionogels do, which decreases the transparency slightly by scattering the light. As shown in Figure 5b, we found that the transparency of SNP-reinforced ionogels showed no obvious change compared with ionogels, indicating that SNPs had little effect on the transparency of ionogels. The SNP-reinforced ionogels were as transparent as the unreinforced ionogels. Transmission spectrum images of blank SNP-reinforced ionogels are shown in Appendix A. An optical microscope photo for 0.5 wt% SNP ionogel shows that SNP aggregates slightly, but there was no aggregation larger than 50 μm observed. Generally speaking, SNPs were evenly dispersed in the ionogels (as shown in Appendix A).

In consideration of their mechanical properties and thermal stability, when the content of crosslinker is 0.8 mol% of monomers, ionogels showed the largest breaking strain. When adding 0.5 wt% SNPs, the breaking stress and the breaking strain were improved at the same time. As the next step, we tested the function of flexible electronic devices constructed from SNP-reinforced ionogels.

### 3.5. Electrodes for Flexible Actuators

In many cases, ionogels are applied as soft electrodes for flexible electro-actuators [3]. The electro-actuation ability of flexible actuators based on SNP-reinforced ionogels is presented in Figure 6. A sinusoidal voltage with a frequency of 0.025 Hz and an increasing amplitude was applied to the flexible actuator. When the sinusoidal voltage was applied, the elastomer covered with SNP-reinforced ionogels displayed expansion and contraction due to the electric field force. When the voltage was off, the elastomer soon recovered to the as-prepared state, as shown in Figure 6a. It expanded quickly when the voltage was on, as shown in Figure 6b. This behavior is quite similar to flexible actuators based on carbon grease, as shown in Figure 6c,d. This result shows that SNP-reinforced ionogels can be applied as soft electrodes for flexible electro-actuators without disturbing their required function.

Further electro-actuation testing was carried out to study the dependency between the actuation ability and voltage. Figure 7 demonstrates the area change rate of the flexible actuators under different voltages. The actuator starts its expansion under a voltage of 1000 V and performs a bigger actuation while at a higher voltage. When the voltage reached 4500 V, the flexible actuator based on SNP-reinforced ionogels was able to expand by about 148.64%. In comparison, the flexible actuator based on carbon grease performed a similar actuation. It expanded to about 147% under 4500 V voltage. This experiment shows quantitatively that the SNP-reinforced ionogels exhibit the same ability with classical carbon grease as soft conductors for dielectric elastomer actuators.

### 3.6. Functional Testing of Stretchable Cables

In order to further evaluate the flexible electro-properties of SNP-reinforced ionogels, a stretchable cable experiment was carried out. Cables (2.5 mm diameter) made from SNP-reinforced ionogel were introduced into an electrical circuit, as shown in Appendix A. Light-emitting diodes work properly with a power supply of 56 V, as can be seen in Figure 8a. When the soft cables based on the SNP-reinforced ionogel were slowly stretched, the light-emitting diodes still worked normally, even under a 400% strain, as shown in Figure 8b. The brightness of the light-emitting diodes was slightly weakened due to the increase of the total resistance of the circuit. Nevertheless, when the cable was restored to its original shape, the light-emitting diodes could recover their original brightness. This experiment can be repeated many times, as can be seen in Appendix A, which shows SNP-reinforced ionogel material exhibiting flexible electro-conductivity.

As a further demonstration, the flexible electro-conductive cable was also applied to the transmission of an electrical signal. The circuit diagram is presented in Appendix A. When the power was on, the loudspeaker worked normally, indicating that the SNP-reinforced ionogel cable did transmit the electrical signal. The volume went down when the cable was stretched, similarly to the performance of the above light-emitting diode experiment. When the cable was restored to its original shape, the volume returned to its original intensity. The repeatability of this experiment was demonstrated through the cyclic stretching of the ionogel cables, which demonstrated the good conductivity and durability of the stretchable cables based on SNP-reinforced ionogels, as shown in Appendix A.

## 4. Conclusions

In conclusion, SNP-reinforced ionogels with various amounts of added SNPs were prepared. Their mechanical, thermal and electrical properties were evaluated and some positive results were obtained. The resultant material is highly stretchable, transparent and nonvolatile, and thus suitable to be applied as a conductor for flexible electronic devices, even when exposed to open air. In this study, ionogels based on [C_2_mim][EtSO_4_] and a polyacrylic network were prepared and the properties of the ionogel with various contents of crosslinking agents and SNP concentrations were studied. Ionogels showed the largest breaking strain when the content of the crosslinker in the ionogels was 0.8 mol% of the monomers. SNPs were also dispersed into the ionogel matrix to reinforce the mechanical properties. The breaking stress and breaking strain were both improved when the content of SNP was 0.5 wt%. A flexible actuator based on SNP-reinforced ionogels was able to expand to about 148.64% when the voltage reached 4500, which is comparable to carbon grease. It is well demonstrated that stretchable cables based on SNP-reinforced ionogels work in a light emitting diode circuit and an audio signal line. In conclusion, SNPs dispersed into the ionogel reinforce its mechanical properties, while maintaining good thermal stability, conductivity and transparency. SNP-reinforced ionogels are expected to be widely applied as large-deformation conductors in the field of flexible electronic devices.

## Figures and Tables

**Figure 1 membranes-10-00354-f001:**
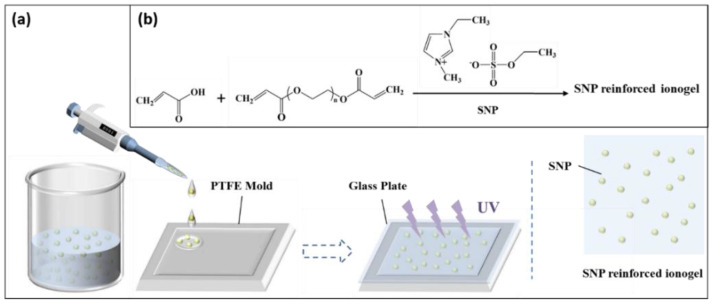
Preparation process of silica nanoparticle (SNP)-reinforced ionogel: (**a**) material preparation and (**b**) chemical reaction process of SNP-reinforced ionogel.

**Figure 2 membranes-10-00354-f002:**
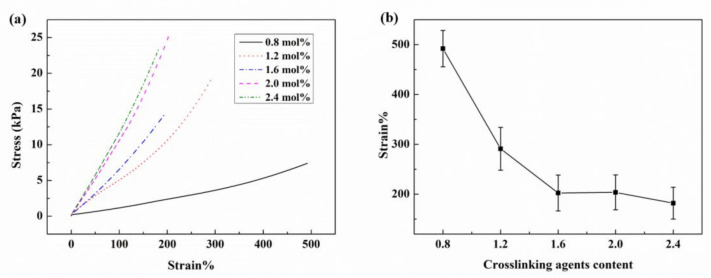
Mechanical properties of ionogel samples with different contents of crosslinking agents: (**a**) stress-strain curves of ionogel samples with different contents of crosslinking agents; (**b**) breaking strain of ionogel samples with different contents of crosslinking agents.

**Figure 3 membranes-10-00354-f003:**
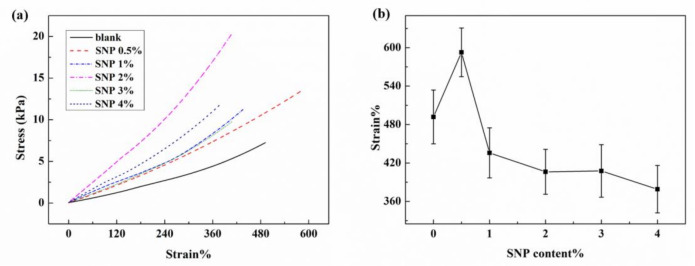
Mechanical properties of ionogels with different contents of SNPs: (**a**) stress-strain curves of ionogels with different contents of SNP; (**b**) breaking strain of ionogels with different contents of SNPs.

**Figure 4 membranes-10-00354-f004:**
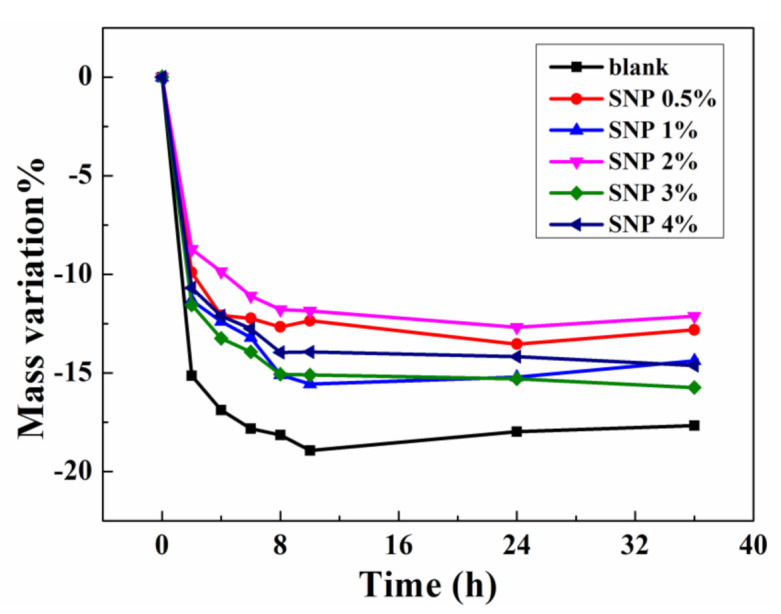
Thermal stability of ionogel samples with different contents of SNPs.

**Figure 5 membranes-10-00354-f005:**
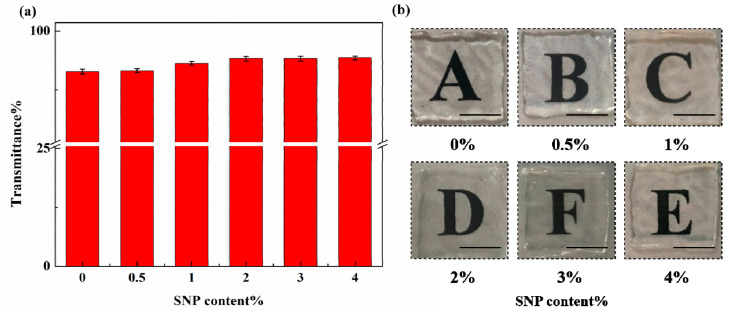
Transparency of SNP-reinforced ionogels with different contents of SNP: (**a**) integral transmittance of ionogels (from 300 nm to 700 nm) with various SNP contents; (**b**) digital photos of ionogels with various SNP contents.

**Figure 6 membranes-10-00354-f006:**
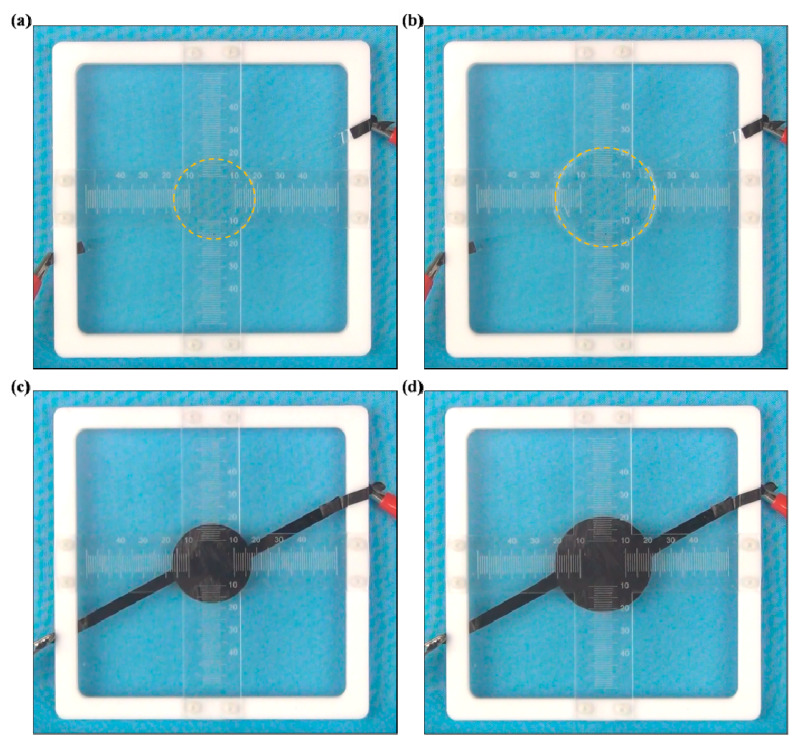
Expansion of the flexible actuator via a sinusoidal voltage: ionogel conductor when the voltage is (**a**) off, (**b**) on; carbon grease conductor when the voltage is (**c**) off, (**d**) on.

**Figure 7 membranes-10-00354-f007:**
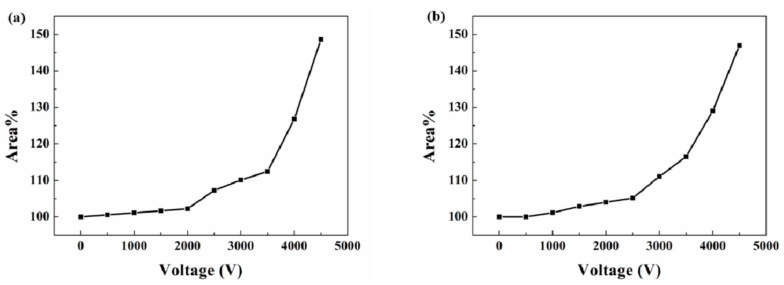
Area change rate of the flexible actuators: (**a**) flexible actuator based on SNP-reinforced ionogels; (**b**) flexible actuator based on carbon grease.

**Figure 8 membranes-10-00354-f008:**
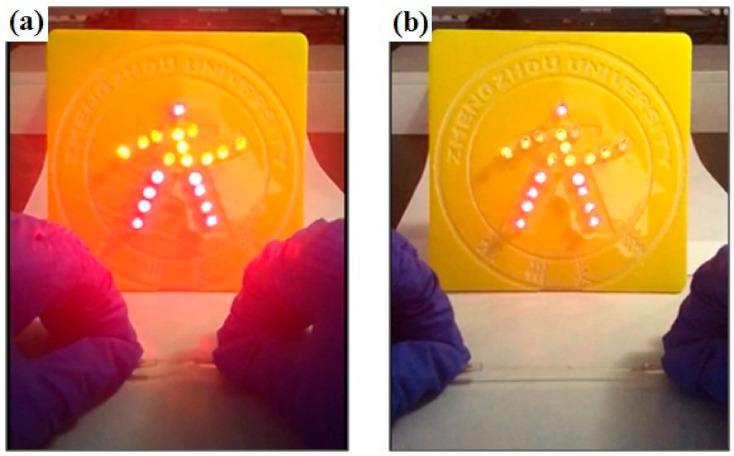
Stretching the stretchable cable based on SNP-reinforced ionogel in a light emitting diode circuit: (**a**) stretchable cable based on SNP-reinforced ionogel without stretching; (**b**) stretching the stretchable cable based on SNP-reinforced ionogel.

**Table 1 membranes-10-00354-t001:** The electroconductivity of ionogels composites.

Sample	Electroconductivity (S/m)
blank	1.28 ± 0.07
SNP 0.5%	1.23 ± 0.07
SNP 1.0%	1.09 ± 0.06
SNP 2.0%	1.11 ± 0.06
SNP 3.0%	1.10 ± 0.06
SNP 4.0%	1.12 ± 0.06

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
