# Peer review of "Silica Nanoparticles Reinforced Ionogel as Nonvolatile and Stretchable Conductors"

_membranes, 2020, doi:10.3390/membranes10110354_

Round 1
Reviewer 1 Report
Comments
The authors provided a concise research on nonvolatile and stretchable conducting ionogels embedded silica nanoparticles, which is the focus of the current research.
While the research topic is timely and will be of general interests to the readerships of Membranes Journal. I have the following criticisms, which the authors should consider during the minor revision of the paper. The paper has well written and following changes should be considered in the manuscript.
- Authors should follow the journal template for texts, fonts, reference citing, etc.
- Authors should provide the SEM image for 0.5 wt% SNP ionogel (To justify the dispersibility of SNP in the ionogel network).
- It seems from Figure S7 and Figure 5, the transparency of ionogels are slightly decreased with SNP content. The reverse order is most preferable (is it any mistake during representations?).
- The plot of thermal stability of ionogel samples with different content of SNP in Figure 4 is not consistent as authors explained in section 3.3. It seems that SNP 2% shows somewhat better stability compared to SNP 0.5% and SNP 4% than SNP 3%. Authors should explain why this phenomenon occurred or repeat the test to get better logical plots.
- There are many typo errors in the manuscript, for example: Page 3, Line 106. Authors has to rectify all the errors after carefully reading the final manuscript.
- What is VHB+ILG in Figure 7a, the abbreviation ILG is not mentioned before in the manuscript.
- It is well known that SNP is a great electrical insulator, how the authors got increased conductivity for SNP content from 1% to 4% (As displayed in Table 1).
Reviewer 2 Report
The manuscript describes the use of silica nanoparticles to reinforce ionogel as stretchable conductors. The article can be further improved by making the following changes in:
1) Improving formatting and grammatical mistakes throughout the manuscript
2) While there is plenty of literature on silica or silica nanoparticles based ionogels, the authors have not cited such research in the introduction which needs to be included in the updated version.
3) Details of experimental methods (UV-vis studies, SEM analysis, resistivity, DMA testing needs to be added in the 'Experimental' section
4) What are the freezing and freeze-drying conditions used on line 96, page 4.
5) How did the authors measure the crosslinking degree of the ionogels?
6) The authors should explain their rationale to synthesize their own silica particles while there is a variety of commercially available silica particles
7) Some properties (e.g., thermal nonvolatility and functional testing) need to be discussed in further details to convey the achievements of the study more effectively
8) What is the variation in the silica particles size? How was that measured?
Round 2
Reviewer 2 Report
The manuscript reads well with clarity of ideas, data, and interpretation after the authors have tried to improve on all the suggestions.
Author Response
Thanks for your comments.